# 1 Quantification of Delayed Recharge by Soil Surface and Riverbed

# 2 Infiltration in a Deep Groundwater Depression Zone in the North

# **China Plain**

- Shenghao Xu<sup>1</sup>, Yonggen Zhang<sup>2\*</sup>, Xinwang Li<sup>3</sup>, Jianzhu Li<sup>1</sup>, Wenhao Shi<sup>2</sup>, Shaowei Lian<sup>4</sup>, Lei Li<sup>5</sup>, Lutz
- Weihermüller<sup>6</sup>, Marcel Schaap<sup>7</sup>
- <sup>1</sup>State Key Laboratory of Hydraulic Engineering Intelligent Construction and Operation, Tianjin University, Tianjin 300350,
- China
- <sup>2</sup>Institute of Surface-Earth System Science, School of Earth System Science, Tianjin University, Tianjin 300072, China
- <sup>3</sup>Hebei Institute of Water Science (Hebei province dam safety technology center, Hebei province levee sluice technology
- center), Shijiazhuang 050051, China
- <sup>4</sup>Hebei Provincial Hydrologic Survey and Research Center, Shijiazhuang 050051, China
- <sup>5</sup>Ninth Geological Brigade of Hebei Bureau of Geology and Mineral Resources, Xingtai 054000, China
- <sup>6</sup>Agrosphere Institute IBG-3, Forschungszentrum Jülich GmbH, Jülich 52428, Germany
- <sup>7</sup>Department of Environmental Science, University of Arizona, Tucson 85721, USA
- Correspondence to: Yonggen Zhang (ygzhang@tju.edu.cn)
- Abstract. Agriculture on the North China Plain (NCP), home to over 300 million people, heavily relies on groundwater 16 extraction to feed its irrigation systems. This has created a large groundwater depression zone up to 80 m deep, severely 17 limiting sustainable groundwater extraction and crop-production. Effective recharge is essential to restore this depleted zone 18 19 and secure future sustainability. Few studies, however, have quantified recharge delays and efficiencies in deep vadose zones 20 with complex lithology. Here we simulated infiltration times and percolation velocities in the Ningbailong groundwater 21 depression zone, a typical overexploited site in the NCP using HYDRUS-1D with measured borehole lithology and hydrometeorological data. Two infiltration modes were considered: precipitation-fed and riverbed infiltration. Spatial distributions 22 23 of infiltration times and percolation velocities were obtained, and recharge efficiencies were compared between these two 24 infiltrations. Results showed that times for precipitation-fed recharge averaged 446 days and varied with lithology and thickness, from 10 days in western Taihang foothills (dominated by coarse sands) to 1,395 days in central/eastern plains (finer 25 clays and loams). Riverbed recharge was markedly faster, averaging 91 days, indicating higher efficiency than precipitation 26 27 under equivalent lithological conditions. Regression equations were derived to predict percolation velocities from vadose zone 28 thickness and soil particle fractions. These findings elucidate how vadose zone thickness and lithology amplify recharge lags and control recharge efficiency. They also highlight the potential for managed aquifer recharge strategies, such as constructing 29

infiltration basins for flood capture, offering strategies to reduce groundwater over-exploitation in similar depression zones.

### 1 Introduction

31

32

Over-exploitation of groundwater resulting in the depletion of aquifers have become pressing global concerns in many regions 34 across the world (Wang et al., 2010; Kuang et al., 2024; Karandish et al., 2025), resulting in geological hazards such as land 35 subsidence, seawater intrusion, and wells running dry (Scanlon et al., 2023; Shirzaei et al., 2021; Jasechko and Perrone, 2021). 36 Groundwater recharge primarily occurs through precipitation-fed soil surface infiltration and riverbed infiltration. In the North 37 China Plain (NCP) which feeds 300 million people, approximately 70% of the shallow groundwater recharge comes from 38 precipitation-fed infiltration through soils (Liu et al., 2022). Quantification of the efficiency of this process in relation to 39 riverbed infiltration as controlled by variability in vadose zone lithologies is fundamental to sustainable utilization and 40 scientific management of depleted groundwater zones. 41 Extensive research has explored the principles of infiltration in the vadose zone based on the theory of unsaturated soil water 42 movement, which governs how water infiltrates through partially saturated porous media under the influences of gravity and matrix potential gradient (Assouline, 2013; Vereecken et al., 2019; Christine and Gerhard, 2022; Schübl et al., 2023; Gao et 43 44 al., 2024). These studies highlight the critical role that heterogeneities in soil lithology (soil texture), structure, and vadose 45 zone depth can play in regulating infiltration rates, lag times, and overall recharge efficiency, particularly in water 46 overexploitation regions (Szabó et al., 2019; Turkeltaub et al., 2015). 47 Among various surface inputs, precipitation-fed infiltration emerges as the primary driver of vadose zone and groundwater recharge, directly linking atmospheric inputs to subsurface hydrology through vertical percolation. For instance, Dafny and 48 49 Šimůnek (2016) calibrated van Genuchten hydraulic parameters for layered loess deposits in Israel's coastal plain using a HYDRUS-2D/3D model informed by infiltration tests, revealing that saturated conductivities varied for different soil 50 51 lithologies. Employing HYDRUS-1D with 25-year meteorological data, they simulated recharge under bare soil, semistabilized dunes, and stabilized landscapes, demonstrating the role of vegetation in reducing recharge through enhanced 52 53 transpiration. Sediment layering caused lag times of 2.5-20 years for wetting fronts to reach 22 m depth, emphasizing lithological control on infiltration efficiency in arid areas. Jie et al. (2022) quantified vadose zone thickness impacts on delayed 54 55 recharge in Jingdian Irrigation District in Northwest China using HYDRUS-1D simulations, showing linear lag increases (up 56 to 5,000 days for depths >8 m) and significantly reduced recharge rates as thickness increases. Extending this to global scales, 57 Moeck et al. (2024) assessed groundwater recharge responses to monthly-decadal infiltration variability using an analytical 58 solution of the Richards equation, finding that vadose zones dampen short-term fluctuations globally, with lags exceeding 59 years in arid regions and transient recharge driven by multi-annual cycles such as ENSO. The correlation between infiltration 60 and recharge weakens with depth, emphasizing how vadose zone thickness and soil properties control infiltration timing and 61 efficiency under variable precipitation regimes. More recently, Yin et al. (2025) analyzed time-varying periods and lags in precipitation-fed recharge from Heilongjiang Basin in China through GRACE data and wavelet transforms, identifying 62

Groundwater recharge is a critical process in the hydrological cycle and a fundamental component in groundwater resources.

63

response under climate change. Riverbed infiltration is another important pathway through which surface water recharges groundwater. Empirical studies have 65 66 demonstrated that this process can also have significant effects on groundwater recharge (Dillon et al., 2019, Niswonger et al., 2005). In regions with low precipitation and high actual evapotranspiration, or in areas with concentrated water inputs from 67 intermittent streams, groundwater recharge is dominated by riverbed infiltration, while precipitation-driven infiltration 68 69 minimally contributes to groundwater recharge (Bierkens et al., 2021). The key factors influencing riverbed infiltration include 70 the material composition of the riverbed, the hydraulic gradient between the river and groundwater, as well as aquifer 71 characteristics (Shanafield et al., 2020). However, due to the lack of direct methods for monitoring the riverbed infiltration, it 72 is typically analyzed through continuous observations of the river water levels, soil water content in the infiltration zone, and 73 groundwater levels. Ruehl et al. (2006) utilized hydrological station data and tracer tests to quantify river leakage rates and 74 assess the reliability of conceptual models and quantitative approaches for studying river leakage processes. Xi et al. (2015) 75 investigated the saturated permeability coefficient of the riverbed in the lower reaches of the Heihe River Basin, Northwest 76 China by using the Guelph Permeameter and laboratory analysis methods and analyzed the characteristics of riverbed 77 infiltration and its spatial distribution patterns. They found that the saturated permeability coefficient exhibited a moderate 78 degree of spatial heterogeneity, with the riverbed material composition, initial soil water content and bulk density having 79 primary impacts on riverbed infiltration. Secondary effects controlling riverbed infiltration included topographic factors such 80 as riverbed width, altitude, hydraulic gradient, and riverbed curvature. Di Ciacca et al. (2024) proposed a model simplification 81 framework that transitions from complex 3D integrated hydrological models to simpler 1D analytical conductance models for 82 simulating groundwater recharge in perched gravel-bed rivers, with application to the Selwyn River in New Zealand. They 83 emphasized the critical influence of riverbed sediment storage and groundwater levels in shallow aquifer on time-variable 84 infiltration rates and recharge volumes under seasonal river conditions. In general, research on groundwater recharge has primarily focused on vertical infiltration into the vadose zone. Widely used 85 methods for evaluating infiltration recharge volumes include physical methods (e.g., Racz et al., 2012; Ganot et al., 2017), 86 87 tracer methods (e.g., Wang et al., 2024), and mathematical models (e.g., Vereecken et al. 2019; Šimůnek et al., 2012; Arnold et al., 2012). Vereecken et al. (2019) provided a comprehensive overview of mathematical models for infiltration processes, 88 89 ranging from empirical approaches such as the Kostiakov equation (1932) and Horton equation (1941) to analytical solutions 90 such as the Green-Ampt (1911) and Philip (1957) models. They emphasized the Richards equation as the fundamental 91 framework for unsaturated flow, incorporating soil hydraulic properties such as the water retention curve  $\theta(h)$  (e.g., Brooks 92 and Corey, 1964; van Genuchten, 1980) and hydraulic conductivity K(h), noting that for real-world infiltration problems (e.g., 93 layered soil profiles, variable initial saturation, time-variable rainfall, and limited ponding), quantitative analysis is typically 94 achieved through numerical solutions of the Richards equation. 95 Leveraging computational advances over the past decades, several software codes have been developed to simulate vadose 96 zone infiltration and groundwater recharge dynamics by numerically solving the Richards equation and related processes.

dominant 1-2 year cycles with lags of 2-6 months in plains, modulated by topography and soil type, highlighting accelerated

97 98

102

128

Well-known methods include HYDRUS (Šimůnek et al., 2012, 2016, 2024), SWAP (Van Dam et al. 2008; Kroes and Supit 2011), and SWMS (Li et al., 2019). Hydrus-1D is a one-dimensional soil water model that comprehensively accounts for precipitation, vegetation water uptake, evaporation, soil water movement, and groundwater table fluctuations (Assefa and Woodbury, 2013; Stafford et al., 2022; Dadgar et al., 2018). Assefa and Woodbury (2013) integrated field observations and HYDRUS-1D to model transient, spatially varied groundwater recharge in North Okanagan, Canada. Coupled with ArcGIS, the model produced recharge maps for the Deep Creek watershed, estimating average recharge at  $77.8 \pm 50.8$  mm year<sup>-1</sup> over 25 years, with significant spatiotemporal variability. Wolf et al. (2022) advanced understanding of recharge mechanisms in 104 thick vadose zones (14-38 m) under irrigated and rangeland land use/land cover, climate variability, and projected climate change to support sustainable groundwater management. Using monitoring data from the High Plains aquifer in central USA, they calibrated HYDRUS-1D models to simulate recharge and total potential profiles for historical (1950-2018) and future 106 107 (1950-2100) periods. Results showed historical recharge lags correlating with the Palmer Drought Severity Index, with land use/land cover as a major control, whereby lag times of 20-24 months at irrigated sites and 5-31 years at rangeland sites were found. Combined these studies show that Hydrus-1D is a valuable tool for quantifying recharge rates and time-delay of deep-110 vadose zone groundwater under a wide spectrum of environmental conditions. The North China Plain (NCP), one of China's three major plains and home to over 300 million people in one of the world's 112 most densely populated regions, is an important area for agricultural production, with groundwater serving as the primary water resource for irrigation (Long et al., 2025). Since the late 1970s, groundwater extraction has intensified in this area, 113 resulting in long term over-exploitation. This has led to the formation of large-scale groundwater depression, making the NCP 114 115 one of the largest groundwater depression zones in the world (Chen et al., 2020). For instance, in the Ningbailong and 116 Gaolisurao regions, where groundwater levels continue to decline, agricultural irrigation became unsustainable (Liu et al., 2022). The substantial decline in groundwater levels has resulted in the formation of a thick vadose zone, affecting water 118 infiltration at the soil surface and groundwater recharge at depth. A holistic understanding soil water movement in these deeper 119 vadose zones is therefore crucial for evaluating recharge mechanisms and developing effective strategies for sustainable 120 groundwater management. Based on field experiments and observations of actual evapotranspiration measured by eddy 121 covariance, Min et al. (2015) used HYDRUS-1D to investigate the vertical infiltration and infiltration characteristics of a thick vadose zone in irrigated farmland in the foothill region of the Taihang Mountains located in the North China Plain. Huo et al. 123 (2014) applied a one-dimensional variably saturated flow model to examine the influence of increasing vadose zone thickness 124 on vertical groundwater recharge in the NCP, and the results showed that as the vadose zone thickens, the magnitude and 125 timing of recharge are significantly altered, with delayed infiltration responses and reduced recharge reaching the water table. 126 Recently, Zhou et al. (2023) quantitatively assessed the sustainability of groundwater in the NCP based on monitoring observations from 556 wells during the period of 2005-2018, using indices such as reliability, vulnerability, and sustainability. They highlighted the weak recovery capacity of the groundwater and identified non-climatic factors as the dominant drivers of depletion in the NCP, emphasizing the profound implications for sustainable water resource management in the region.

131132

135136

137138

139140

142143

148149

Despite the fact that there have been substantial advances in the quantification of precipitation-fed groundwater recharge, few efforts have addressed the challenges in areas with deepening groundwater levels and complex vadose zone lithology. In such areas, groundwater recharge takes much longer due to delayed water percolation through thick variable unsaturated zones, particularly overlooking the critical roles of increasing lag times and average percolation velocities. Existing studies have analyzed precipitation infiltration and riverbed infiltration independently, with no efforts undertaken to systematically compare their efficiencies under identical vadose zone conditions, particularly in terms of recharge times and rates across heterogeneous vadose zone. As one of the world's largest groundwater depression zones, the NCP serves as a typical region for studying infiltration dynamics under intensive overexploitation, where incorporating actual vadose zone borehole data and hydrogeological conditions into quantitative analyses of infiltration times and average percolation velocities can elucidate recharge efficiencies of different sources, including precipitation and riverbed infiltration. This approach also offers insights into spatially heterogeneous groundwater replenishment dynamics across various regions. The main objectives of this paper are to: (1) quantitatively assess the groundwater recharge times and percolation velocities under two recharge regimes (i.e., precipitation-fed infiltration and riverbed infiltration), using measured borehole lithological data and hydrometeorological observations from the Ningbailong groundwater depression zone; (2) analyze the spatial distribution of infiltration times and percolation velocities across the region, accounting for vadose zone heterogeneity influenced by thickness and lithology; and (3) compare the recharge efficiencies of precipitation and riverbed infiltration sources under equivalent vadose zone conditions, provide the empirical regression equation for the percolation velocities under the two recharge regimes, and propose appropriate groundwater recharge methods at the corresponding locations based on the comparison results. The results of this study are anticipated to provide a basis for assessing groundwater over-exploitation and developing management measures in similar depression zones.

# 2 Materials and methods

# 151 **2.1 Study area**

The Ningbailong shallow groundwater depression zone is located in Xingtai City, Hebei Province, China, within one of the 152 main grain-producing regions in the NCP, as shown in Figure 1. The area of the depression zone is 2092 km<sup>2</sup>, with the main 153 river within it covering approximately 7.2 km<sup>2</sup>, accounting for 0.35% of the total area. This area spans the western part of 154 155 Ningjin County, most of Baixiang County, and the northern to central part of Longyao County and it is expanding to the 156 southern part of Shijiazhuang city. The region lies within the Taihang Mountain foothill plain, characterized by a temperate semi-humid to semi-arid continental monsoon climate. The study area borders the Taihang Mountains to the southwest, adjoins 157 Shijiazhuang to the northwest, connects with Hengshui to the northeast, and is adjacent to Xingtai counties to the south and 158 southeast. The topography of the Ningbailong depression zone slopes downward from west to east, with elevations decreasing 159 160 from approximately 100 m in the western foothills to around 30 m in the eastern and northern regions, shown in Figure 1b. Due to its location near the Taihang Mountains, the surface of the western part of the depression zone has steeper slopes (1.5– 161

2.5‰) than the northeastern part (0.5–1‰). Groundwater recharge in the Ningbailong depression zone primarily originates from precipitation, irrigation, and lateral runoff, with precipitation contributing approximately 70% of total recharge. Recharge dynamics are modulated by vadose zone lithology, groundwater depth, topography, and vegetation, as evidenced by regional hydrogeological studies (Cao et al., 2016; Min et al., 2019). Notably, the foothill alluvial fans and eastern plain riverbed zones may have enhanced recharge due to their permeable sediments and close connection to surface water inputs. Since the late 1970s, escalating agricultural water demand has intensified groundwater extraction in the region. Consequently, the Ningbailong groundwater depression zone has expanded in both depth and extent, rendering agricultural irrigation increasingly unsustainable due to persistent groundwater level declines and an uneconomical rise in extraction costs.

Figure 1: Study area of the Ningbailong groundwater depression zone in the North China Plain with (a) location of the Ningbailong groundwater depression zone, (b) topography, and (c) groundwater depth with groundwater monitoring wells as blue dots.

### 2.2 Data

Hydrogeological data for the Ningbailong depression zone were sourced from the 9<sup>th</sup> Geological Brigade of the Hebei Provincial Bureau of Geological and Mineral Exploration and Development, encompassing borehole logs, water supply wells, pumping test results, geophysical prospecting data, and groundwater level contour maps. Since 2018, dynamic groundwater level monitoring has provided continuous data to characterize temporal variability in the depression zone. Vadose zone profile data, including lithological type, soil water content, and vadose zone thickness, were collected from multiple sampling sites across the region to assess vadose zone heterogeneity. The lithology of the study area was analyzed and conceptualized using borehole logs for 24 groundwater wells (see Table 1). In Table 1, the "Infiltration Mode" column uses "P" or "R" to respectively indicate that the well is currently receiving precipitation-fed or riverbed infiltration.

The meteorological and hydrological data for the study area were provided by the Xingtai Hydrological Survey and Research Center of Hebei Province. These data include daily precipitation, potential evapotranspiration, river water level records from

Center of Hebei Province. These data include daily precipitation, potential evapotranspiration, river water level records from the local precipitation stations and hydrological stations since 2014. These datasets served as driving data for simulating the two infiltration scenarios (precipitation-fed soil infiltration and riverbed recharge).

Table 1: Representative borehole data in the Ningbailong groundwater depression zone in the North China Plain. The infiltration mode column specifies the recharge mode for each borehole, where "P" represents precipitation-fed infiltration and "R" represents riverbed infiltration.

| <b>Borehole Name</b> | Infiltration Mode | Location               | Depth (cm) | Latitude(°) | Longitude(°) |
|----------------------|-------------------|------------------------|------------|-------------|--------------|
| Bai 1                | P                 | Baixiang, Xingtai      | 4960       | 37.6123     | 114.7186     |
| Bai 10               | P                 | Baixiang, Xingtai      | 6720       | 37.5434     | 114.7301     |
| Bai 11               | R                 | Baixiang, Xingtai      | 5080       | 37.4845     | 114.6427     |
| Bai 12               | P                 | Baixiang, Xingtai      | 8080       | 37.5787     | 114.7110     |
| Bai 18               | P                 | Baixiang, Xingtai      | 6150       | 37.4729     | 114.6828     |
| Xisucun              | P                 | Baixiang, Xingtai      | 4060       | 37.5992     | 114.7671     |
| Xiaonanyangcun       | P                 | Baixiang, Xingtai      | 5750       | 37.4777     | 114.7484     |
| Hancun               | R                 | Baixiang, Xingtai      | 5120       | 37.5501     | 114.6109     |
| Ning 17              | P                 | Ningjin, Xingtai       | 5960       | 37.6709     | 114.8331     |
| San 62               | P                 | Ningjin, Xingtai       | 5170       | 37.6913     | 115.1836     |
| Ning 18              | P                 | Ningjin, Xingtai       | 4600       | 37.5444     | 115.0280     |
| Ning 20              | P                 | Ningjin, Xingtai       | 3150       | 37.6873     | 115.0348     |
| ZK 1                 | P                 | Ningjin, Xingtai       | 4200       | 37.5238     | 114.9289     |
| Guoce 510            | R                 | Longyao, Xingtai       | 3620       | 37.3460     | 114.7390     |
| Long 8               | P                 | Longyao, Xingtai       | 5960       | 37.5086     | 114.8314     |
| Long 12              | P                 | Longyao, Xingtai       | 2130       | 37.3108     | 114.6725     |
| Gao 1                | P                 | Gaoyi, Shijiazhuang    | 5640       | 37.6100     | 114.6611     |
| Zhao 1               | P                 | Zhaoxian, Shijiazhuang | 5290       | 37.6836     | 114.7588     |
| Zhao 2               | P                 | Zhaoxian, Shijiazhuang | 5350       | 37.7585     | 114.7415     |
| CK 18                | P                 | Zhaoxian, Shijiazhuang | 6560       | 37.7648     | 114.9168     |

| <b>Borehole Name</b> | Infiltration Mode | Location               | Depth (cm) | Latitude(°) | Longitude(°) |
|----------------------|-------------------|------------------------|------------|-------------|--------------|
| CK 21                | P                 | Jinzhou, shijiazhuang  | 5700       | 37.9000     | 115.1211     |
| CK 22                | P                 | Jinzhou, shijiazhuang  | 5410       | 37.8541     | 115.0827     |
| CK 3                 | P                 | Gaocheng, shijiazhuang | 4860       | 37.9298     | 114.9478     |
| CK 10                | P                 | Gaocheng, shijiazhuang | 4660       | 37.8936     | 114.8191     |

### 2.3 Simulation setup

- This study employed the HYDRUS-1D model (Šimůnek et al., 2012, 2024) to simulate one-dimensional soil water movement
- in the unsaturated zone by numerically solving the Richards equation. Widely applied in groundwater recharge investigations,
- HYDRUS-1D effectively captures soil water dynamics under varying recharge scenarios in the vadose zone.
- This study considered only one-dimensional vertical flow, neglecting horizontal overland flow and lateral movement of soil
- water in the vadose zone. The ground surface is taken as the origin of the coordinate system, with the positive direction of the
- z-axis pointing downward. The Richards equation for one-dimensional saturated-unsaturated zone soil water movement is
- given by:

189

$$\frac{\partial \theta}{\partial t} = \frac{\partial}{\partial z} \left[ K(\theta) \left( \frac{\partial h}{\partial z} \right) + K(\theta) \right] - S(z, t) \tag{1}$$

- where  $\theta$  is the volumetric soil water content (cm<sup>3</sup> cm<sup>-3</sup>), t is represents time (d), z represents the soil depth (cm),  $K(\theta)$  indicates
- the unsaturated hydraulic conductivity (cm d-1) with respect to water content, h represents the pressure head (cm), which is
- $h \ge 0$  in the saturated zone and h

Figure 2: USDA soil types and stratification for representative borehole in the study area.

## 215 2.3.2 Soil hydraulic parameters

213

The soil hydraulic properties were described by the modified van Genuchten model (van Genuchten, 1980; Vogel et al., 2000).

The soil water retention characteristic  $\theta(h)$  and hydraulic conductivity  $K(\theta)$  are given by:

$$\theta(h) = \begin{cases} \theta_a + \frac{\theta_m - \theta_a}{\left(1 + \left|\alpha h\right|^n\right)^m} & h < 0\\ \theta_s & h \ge 0 \end{cases}$$
 (2)

$$K(h) = \begin{cases} K_s K_r(h) & h < 0 \\ K_s + \frac{(h - h_k)(K_s - K_k)}{h_s \times h_k} & h_k < h < 0 \\ K_s & h > 0 \end{cases}$$
 (3)

$$K_{r} = \frac{K_{k}}{K_{s}} \left( \frac{S_{e}}{S_{ek}} \right) \left[ \frac{F(\theta_{r}) - F(\theta)}{F(\theta_{r}) - F(\theta_{kr})} \right]^{2}$$
(4)

$$F(\theta) = \left[1 - \left(\frac{\theta - \theta_{\alpha}}{\theta_{m} - \theta_{\alpha}}\right)^{1/m}\right]^{m}$$
 (5)

$$S_{ek} = \frac{\theta_k - \theta_r}{\theta_r - \theta_r} \tag{6}$$

where  $\theta_r$  represents the residual soil water content (cm<sup>3</sup> cm<sup>-3</sup>),  $\theta_s$  indicates the saturated water content (cm<sup>3</sup> cm<sup>-3</sup>),  $\alpha$ , n, and m are empirical parameters, whereby m can be related to n by m=1-1/n,  $K_s$  is the saturated hydraulic conductivity (cm d<sup>-1</sup>),  $K_k$  represents the unsaturated hydraulic conductivity (cm d<sup>-1</sup>) at the pressure head  $h_k$ ,  $K_r$  represents the relative hydraulic conductivity (-), and  $S_e$  is the effective saturation (-). Based on borehole measurements and referring to relevant literature (Weihermüller et al., 2021), we adopted the improved hierarchical pedotransfer function set (Rosetta3) developed by Zhang and Schaap (2017) to derive the van Genuchten parameters and  $K_s$  as the soil hydraulic parameter values of the unsaturated zone. The resulting values are presented in Table 2.

Table 2. Soil hydraulic parameters used for vadose zone modelling in this study.

| Soil type       | $\theta_r (\text{cm}^3 \text{cm}^{-3})$ | $\theta_s$ (cm <sup>3</sup> cm <sup>-3</sup> ) | α (cm <sup>-1</sup> ) | n (-) | $K_s$ (cm d <sup>-1</sup> ) |
|-----------------|-----------------------------------------|------------------------------------------------|-----------------------|-------|-----------------------------|
| sand            | 0.05                                    | 0.43                                           | 0.145                 | 2.68  | 712.8                       |
| sandy loam      | 0.06                                    | 0.41                                           | 0.075                 | 1.89  | 106.1                       |
| silt loam       | 0.07                                    | 0.45                                           | 0.020                 | 1.41  | 10.8                        |
| sandy clay loam | 0.10                                    | 0.39                                           | 0.059                 | 1.48  | 31.4                        |
| clay loam       | 0.09                                    | 0.41                                           | 0.019                 | 1.35  | 6.24                        |
| sandy clay      | 0.10                                    | 0.38                                           | 0.027                 | 1.35  | 2.88                        |
| clay            | 0.07                                    | 0.38                                           | 0.008                 | 1.31  | 4.8                         |

#### 2.3.3 Generalization of boundary conditions

The simulations were performed for a one-dimensional soil column under vegetated conditions, without accounting for lateral flow, whereby the water movement was simulated under different recharge source types, i.e., precipitation or riverbed infiltration. For the precipitation case, atmospheric boundary conditions were applied at the surface, allowing ponding at the surface if precipitation exceeded soil infiltration capacity, with the minimum allowable surface pressure head (hCritA) set to its default value of -100,000 cm. Excess ponded water was allowed to infiltrate at times where the soils allowed infiltration. In contrast, the riverbed infiltration case employed variable pressure heads at the upper boundary to represent surface water inputs. Because groundwater levels fluctuate over time and depend on net infiltration from surface sources, dynamic water level variations were not assumed to maintain model simplicity. Instead, a constant pressure head condition was imposed at the lower boundary for both scenarios, with the long-term average groundwater level serving as a reference depth for calculating groundwater infiltration times and rates. Figure 3 illustrates the conceptualized vadose zone model setup for the boundary conditions under the two recharge regimes. Precipitation inputs were derived from measured data at the Baixiang Rain Gauge Station, spanning July 2016 to July 2023. The river water level data were obtained from the hydrological stations near the Bai 11, Hancun, and Guoce 510 boreholes during the flood season from July to August 2016, and then apply the same average water level data from these stations to simulate the riverbed infiltration to all the boreholes that are not close to the river.

Lower boundary condition (constant pressure head)

Upper boundary condition under:

- 1 precipitation infiltration (atmospheric boundary condition with surface layer).
- ② riverbed infiltration (variable pressure head)
- Figure 3: Schematic diagram of soil column boundary generalization used in the HYDRUS-1D simulations for the simulation of precipitation infiltration and riverbed infiltration.
- 2.3.4 Root water uptake
- Root water uptake was simulated only for the precipitation-fed infiltration scenario, as vegetation is typically absent in
- riverbeds, rendering transpiration negligible in those cases. For root water uptake, the Feddes model (Wesseling and Feddes,
- 2006) was used, which is expressed by:

$$S(z,t) = a_f(h)\gamma(z)T_p \tag{7}$$

$$T_{p} = ET_{0}(1 - e^{-kLAI})$$
 (8)

where  $a_f(h)$  represents the water stress function ( $0 \le a_f \le 1$ ), which reflects the reduction in root water uptake due to soil water depletion. Based on the conditions in the study area, the root water uptake model parameters were selected from the "Corn (Wesseling, 1991)" option provided by the HYDRUS-1D, as detailed in Table 3.  $\gamma(z)$  is the root water uptake distribution function (cm<sup>-1</sup>), which reflects the spatial variability of root water uptake within the vertical soil profile. In this study, a linearly

decreasing function was adopted, with a maximum rooting depth of 100 cm.  $T_p$  represents the potential transpiration rate (cm d<sup>-1</sup>).  $ET_0$  represents the reference evapotranspiration (cm), which can be estimated using the FAO-recommended Penman-Monteith method (Pereira et al., 2015). LAI represents the leaf area index (dimensionless), obtained from values of vegetation LAI taken from Zhang et al. (2015). k represents the radiation extinction coefficient of the plant canopy (dimensionless), and is set to the default value of 0.4.

Table 3. Feddes root water uptake parameters.

| P <sub>0</sub> (cm) | P <sub>0pt</sub> (cm) | P <sub>2H</sub> (cm) | P <sub>2L</sub> (cm) | P <sub>3</sub> (cm) | r <sub>2H</sub> (-) | r <sub>2L</sub> (-) |
|---------------------|-----------------------|----------------------|----------------------|---------------------|---------------------|---------------------|
| -15                 | -30                   | -325                 | -600                 | -8000               | 0.5                 | 0.1                 |

### 2.3.5 Model spin-up

The initial distribution of soil water content throughout the vadose zone depth is essential for obtaining reliable simulation results, as it directly influences simulated infiltration rates and actual recharge to groundwater. However, due to the considerable thickness of the vadose zone, it is not possible to provide measured initial soil water content profiles. To address this limitation, a sufficiently long spin-up period was incorporated to equilibrate soil water contents along the entire profile (Jie et al., 2022). Accordingly, the model spin-up spanned July 2016 to July 2022, after which groundwater recharge analysis started from August 1, 2022. An initial pressure head of h = -50 cm was uniformly assigned to the unsaturated zone as a predefined condition at the beginning of the spin-up.

### 2.4 Spatial interpolation

and coordinate transformation to produce a sample distribution map suitable for geostatistical analysis. The corresponding groundwater level data for each point were integrated with the geographic attributes to create comprehensive datasets. In the simulations, infiltration was deemed to have reached the profile base when outflow occurred at the bottom boundary. This timestamp defined the infiltration time of percolating water through the vadose zone. To allow fair comparison between different vadose zone thicknesses, the normalized infiltration time (average percolation velocity) was computed as the ratio of vadose zone thickness to infiltration time.

To obtain the infiltration time and rate distributions across the entire study area from the simulated 24 individual point locations by HYDRUS-1D, the Inverse Distance Weighting (IDW) method was employed for spatial interpolation of the borehole data. This approach estimates values at unsampled locations using weights inversely proportional to distances from known points, calculated as:

Using QGIS, the locations of the 24 shallow observation wells were converted into point shapefiles, followed by projection

$$Z(x) = \frac{\sum_{i=1}^{n} \frac{Z_i}{d_i^p}}{\sum_{i=1}^{n} \frac{1}{d_i^p}}$$
 (9)

- where Z(x) represents the value at the unknown point to be interpolated,  $Z_i$  indicates the value at the known point,  $d_i$  is the
- distance between the unknown point and the known point, and p denotes the weighting factor which is set to 2 by default.

### 2.5 Multiple regression analysis

- To investigate the influence of vadose zone thickness and lithological characteristic on groundwater recharge efficiency, we
- modeled percolation velocities (PV, cm d<sup>-1</sup>) as the dependent variable, which directly quantifies the rate of water movement
- through the vadose zone and thus serves as a key indicator of recharge efficiency. Three key factors, vadose zone thickness
- (m), clay fraction (-), and sand fraction (-) were selected as independent variables to establish the relationships influencing PV.
- To account for skewed distributions and improve model fit, the natural logarithm was applied to PV and vadose zone thickness.
- A leave-one-out cross-validation procedure was implemented using R code. In this procedure, each observation in the dataset
- (num = 24) was sequentially held out as the validation set, while the remaining num-1 observations were used for regression
- model calibration.
- Within each iteration, a multiple regression model incorporating main effects and pairwise interaction terms was fitted. Model
- parameters were estimated using the ordinary least squares method. The regression coefficients obtained from each leave-one-
- out cross-validation iteration are provided in Appendix A, along with the coefficient of determination  $(R^2)$  for the training set
- in each iteration. Overall model performance was evaluated by aggregating predictions across all iterations to compute a
- validation R<sup>2</sup> on the full dataset. Separate regression models were developed for precipitation-fed recharge (Table A.1) and
- riverbed recharge (Table A.2) conditions.

### **304 3 Results**

# 3.1 Validation of soil hydraulic parameters

- To validate the soil hydraulic parametrization with Rosetta3 (Zhang and Schaap, 2017), we compared simulated soil water
- contents with measured data from the Luancheng Agricultural Ecosystem Experimental Station (Wu et al., 2023), which is
- situated within the study area. This station, representative of typical irrigated farmland on the NCP, features a 48-m-deep
- vadose zone observation caisson that enables continuous monitoring of soil water content and matric potential throughout the
- profile.
- For this validation, we configured the HYDRUS-1D model using the soil profile data from this station, applying the same
- model setting strategy as described. Soil water content profiles at this site were measured from June to October 2021 with an
- interval of 1 day to 15 days, totaling 20 dates, and the measurement depth spans from 0 to 44 m. To evaluate model performance,
- we compared simulated soil water content at intervals of 1 m (i.e., 1, 2...44 m) across all 20 measurement dates with the
- corresponding observed values. As shown in Figure 4, the simulated soil water contents closely match the observed values
- across these depths, demonstrating strong model performance ( $R^2 = 0.71$ , RMSE = 0.06 cm<sup>3</sup> cm<sup>-3</sup>). This agreement confirms

the suitability of the Rosetta3-derived parameters for simulating vadose zone dynamics in the study region, providing a reliable soil hydraulic parameters for the subsequent recharge analyses.

Figure 4: Relationship between measured and simulated daily soil water content (cm<sup>3</sup> cm<sup>-3</sup>) at Luancheng Station, North China Plain. The black dashed line represents the 1:1 relationship line.

## 3.2 Impact of vadose zone lithology and thickness on recharge under precipitation-fed conditions

### 3.2.1 Infiltration times and average percolation velocities under precipitation recharge

Simulations of the infiltration process under precipitation recharge conditions using the HYDRUS-1D model yielded estimates of groundwater recharge times and average percolation velocities at borehole locations across the study area (Figure 5). These results indicated substantial variability, with an average infiltration time of 446 days across all sites. The maximum value reached 1,395 days at borehole CK21, while the minimum was only 10 days at borehole Long 12. Infiltration times under this recharge scenario were primarily governed by vadose zone thickness and soil lithology, reflecting the interplay between these factors in controlling water movement through the unsaturated zone. For instance, thicker vadose zones were associated with longer infiltration times, as observed at CK21 and CK18. Even at comparable thicknesses, sites dominated by finer-textured soils such as loam or clay (e.g., Zhao1 and ZK1) exhibited extended infiltration times. In contrast, regions with coarser textures and higher saturated water contents, particularly in the piedmont zone of the Taihang Mountains, facilitated faster percolation. This was evident at Long 12 in Longyao County, where infiltration reached the lower boundary in just 10 days.

A clear inverse relationship emerged between infiltration time (delay) and average percolation velocity, highlighting how lithological and vadose zone thickness propagate to dictate recharge dynamics, i.e., longer delays inherently correspond to slower velocities, as water encounters greater resistance in finer or thicker profiles. Corresponding average percolation velocities further highlighted these lithological influences, averaging 26.4 cm  $d^{-1}$  across the sites, with a maximum of 213.0 cm  $d^{-1}$  at Long 12 and a minimum of 4.1 cm  $d^{-1}$  at CK21. This inverse pattern was particularly pronounced in heterogeneous zones, where coarse-grained lithologies accelerated flow (high velocity, short delay) while fine-grained or layered sections impeded it (low velocity, long delay), emphasizing the need to account for such variability in recharge predictions. To visualize the temporal progression of recharge, Figure 6 illustrates the depth of the infiltration wetting front at selected simulation times under the precipitation scenario. By day 50 (Figure 6a), the front had advanced beyond 20 m at most locations, with the most rapid progression occurring in Baixiang County, where it reached approximately 40 m. By day 800 (Figure 6c), the wetting front had typically reached the base of the vadose zone across the study area, indicating that sustained precipitation inputs eventually overcome lithological barriers, albeit with significant lags in finer-textured or thicker profiles. These patterns emphasize how vadose zone heterogeneity modulates recharge efficiency, with implications for the spatial distribution of groundwater recovery in the Ningbailong depression zone.

Figure 5: Groundwater recharge time (day) and average percolation velocity (cm  $d^{-1}$ ) for locations under precipitation infiltration recharge scenarios.

Figure 6: Depth maps of the infiltration front under the precipitation infiltration recharge scenario at (a) 50, (b) 200, (c) 800, and (d) 1600 days.

Based on the simulation results, a multiple regression model was developed using borehole data to investigate the logarithm of the average percolation velocity under precipitation recharge conditions. The regression functions obtained through leave-one-out cross-validation were provided in Appendix A, including the regression coefficients for each variable and the corresponding  $R^2$  for each calibration set. The  $R^2$  was calculated based on the squared Pearson correlation coefficient between observed and estimated values, which emphasizes the strength of the linear association. As shown in Table A.1, the  $R^2$  values for the calibration range from 0.52 to 0.83, indicating that the model structure is suitable for the data. The overall validation  $R^2$ , computed as the squared correlation between observed and estimated values across all iterations, is 0.47. Due to this small number of samples, the validation  $R^2$  is relatively modest, as limited data can introduce some uncertainty in the model's

367368

371372

374375

generalizability. For a better understanding of the parameter consistency, Table A.1 also includes the average values and standard deviations of each regression coefficient across the calibration sets. This regression model offers significant practical utility for groundwater management and assessments in the Ningbailong

This regression model offers significant practical utility for groundwater management and assessments in the Ningbailong depression zone and similar overexploited regions. By incorporating site-specific inputs for vadose zone thickness, clay content, and sand content into the equations, water resource policymakers can predict average percolation velocities at unsampled locations. Since infiltration time is inversely related to percolation velocity, these predictions enable estimates of recharge delays, which are essential for understanding the lag between surface inputs (e.g., precipitation events) and actual groundwater replenishment. For example, in areas with thick vadose zones and high clay content—common in the central and eastern plains—the model consistently shows negative coefficients for Depth and interactions such as Depth\*Clay and Depth\*Sand, indicating an overall trend toward slower percolation velocity and prolonged delays (potentially exceeding 1,000 days) as these factors increase. Additionally, the inclusion of quadratic terms, such as positive Depth² and Clay², captures nonlinear effects, implying that while thickness and clay initially slow percolation strongly, this impact weakens (i.e., the rate of slowing decreases) at higher values, potentially indicating the slowing impact of greater thickness and clay content becomes weaker at higher levels of vadose zone thickness or clay content. While the model's predictive power is constrained by the small dataset,

# 3.2.2 Spatial distribution of infiltration times and average percolation velocities

To extend the point-scale simulations to a regional perspective, we applied the inverse distance weighting (IDW) interpolation 378 379 method to the HYDRUS-1D results from the 24 boreholes, generating spatial distribution maps of infiltration times and average 380 percolation velocities across the Ningbailong depression zone (Figure 7). These interpolated patterns revealed pronounced 381 spatial variability in groundwater recharge dynamics, driven primarily by differences in vadose zone thickness and lithology. 382 Overall, the average infiltration time in the study area is 463 days, and the average percolation velocity is 26.7 cm d<sup>-1</sup>. The infiltration times shown in Figure 7a increased progressively from the southwestern foothills of the Taihang Mountains toward 383 the central and northern regions of the study area, forming a distinct low-to-high infiltration time gradient that mirrored 384 385 topographic and geological transitions. 386

it captures key lithological controls on recharge dynamics, providing an empirical tool for preliminary assessments.

In the Taihang Mountain foothill region, exemplified by the Long 12 borehole in Longyao County, shallow groundwater depths combined with coarse-grained sand and gravel lithologies in the vadose zone enabled rapid percolation, with infiltration reaching the groundwater in only 10 days. Similarly, the southern sector of Longyao County exhibited infiltration times generally below 100 days, corresponding to average percolation velocities exceeding 100 cm d<sup>-1</sup>. These zones typically coincided with locations near riverbeds or thinner vadose zones, where high-porosity and permeable soils promoted efficient downward water movement and enhanced recharge potential.

In contrast, infiltration times shown in Figure 7a lengthened to 200-600 days, accounting for most of the study are, including Lincheng, Baixiang, Gaoyi, Ningjin, Gaocheng, and parts of Neiqiu, Longyao, and Zhaoxian, where average percolation velocities were less than 60 cm d<sup>-1</sup>. This intermediate regime reflected a shift toward moderately thicker vadose zones and

finer soil textures, which reduced percolation velocities. The longest infiltration times occurred in the central and northern areas, encompassing the boundary of Jinzhou and Xinji, and eastern Zhaoxian (see the red area in Figure 7a), where durations often exceeded 1,000 days and average percolation velocities dropped below 20 cm d<sup>-1</sup>. Such delays were attributable to thick vadose zones and low-permeability soils, which collectively slowed water flux and resulted in extended recharge. Consequently, these regions experience diminished recharge efficiency, limiting timely contributions to groundwater recovery in the depression zone.

Figure 7: (a) Infiltration times distribution under precipitation infiltration conditions, and (b) average percolation velocity distribution under precipitation infiltration conditions.

# 3.3 Impact of recharge sources under riverbed infiltration and comparison with precipitation-fed recharge

To compare recharge efficiencies, we examine infiltration under riverbed conditions using the same vadose zone profiles in this section, quantifying times and velocities for direct comparison with precipitation-fed infiltration. Figure 8 illustrates the groundwater recharge time and average percolation velocity at borehole locations across the study area under riverbed recharge conditions. The average infiltration time across all sites was 91 days, with the longest value observed at CK 18 (268 days) in Zhaoxian County and the shortest at Long 12 (10 days) in Longyao County. Similar to the precipitation-fed recharge scenario, infiltration time in this case was also influenced by vadose zone thickness and soil lithological composition, with an inverse relationship observed between velocity and lag time. The average percolation velocity across the sites was 109.1 cm d<sup>-1</sup>, ranging from a minimum of 24.5 cm d<sup>-1</sup> at CK 18 in Zhaoxian County to a maximum of 397.3 cm d<sup>-1</sup> at Ning 17 in Ningjin County.

Figure 8: Groundwater recharge time (day) and average percolation velocity (cm d-1) for locations under riverbed infiltration recharge scenarios.

Based on the simulation results, a multiple regression model was developed using borehole data to characterize the average percolation velocity under riverbed recharge conditions, following the same methodology as for the precipitation-fed model. The estimated regression coefficients, as well as the average values and standard deviations of the coefficients, are listed in Table A.2, with calibration  $R^2$  values ranging from 0.76 to 0.87, indicating a good fit comparable to the precipitation model. Unlike the precipitation-fed model, the regression equation does not include quadratic terms for depth or clay (i.e., no depth<sup>2</sup> or clay<sup>2</sup> terms), as we found that removing these terms improved the fitting to the regression curves. The overall validation  $R^2$ 

within the depression zone.

(squared correlation between observed and predicted values) is 0.47, affirming the model's utility despite the small sample 423 424 size, though this modest value underscores the need for larger datasets to prevent the overfitting and reduce prediction 425 variability. 426 Similar to the precipitation model, this regression work enables rapid estimation of percolation velocities, and thus recharge 427 delays, using vadose zone thickness, clay content, and sand content, but it reveals distinct dynamics under riverbed conditions. For instance, Table A.2 shows consistently positive coefficients for the main effect of depth (averaging around 1.44 across 428 429 folds), contrasting with the negative coefficients in Table A.1 (averaging around -15.94), indicating that thicker vadose zones 430 facilitate higher log-transformed velocities under constant-head infiltration. This aligns with the observed higher average 431 velocities (109.1 cm d<sup>-1</sup> vs. 26.4 cm d<sup>-1</sup> for precipitation), implying shorter delays even in clay-rich zones, potentially by 4–5 times based on coefficient magnitudes from the log transformation. Such patterns highlight riverbed recharge's resilience to 432 433 lithological barriers, making this approach particularly valuable for optimizing managed aquifer recharge (MAR) strategies, e.g., identifying appropriate sites where artificial riverbed-like basins could accelerate recovery. Future enhancements with 434 more boreholes could refine these insights, enabling probabilistic delay forecasting under varying flood scenarios. 435 436 In order to analyze the impact of different recharge sources on groundwater recharge, we compared the infiltration times and average percolation velocities at the same boreholes under two infiltration conditions, as shown in Figure 9. These comparisons 437 438 enabled to assess the recharge efficiency and highlight the differences in recharge time and spatial scope between the two 439 conditions. Overall, the infiltration duration of the riverbed was shorter and more efficient. However, there were also certain differences based on location. For example, in the southwestern part of the Ningbailong groundwater depression zone, at the 440 Long 12, Hancun, and Bai 10 in Figure 9b, the average infiltration velocity of precipitation and riverbed recharge were similar. 441 442 While in most of the depression areas, the central and northern parts, the differences in the two infiltration recharge rates were 443 significant, such as Ning 17, Bai 1, and Ning 20 shown in Figure 9b. 444 In general, riverbed recharge exerted a stronger influence on groundwater, particularly in localized areas near channels, although its overall spatial footprint remained constrained. Precipitation recharge, by contrast, induced slower groundwater 445 recharge but contributed over a broader regional scale, highlighting its role in widespread, albeit delayed, groundwater recovery 446

Figure 9: Comparison of (a) infiltration time and (b) percolation velocities between two recharge sources within the study area.

### 4 Discussion

449

453454

In this study, we quantified infiltration times and percolation velocity through the vadose zone at boreholes within the Ningbailong groundwater depression zone, using measured borehole data and hydrometeorological records starting from July 1, 2022. This approach enabled the quantification of recharge efficiency by precipitation-fed infiltration through soils and riverbed infiltration. The findings demonstrate that the development of the Ningbailong depression zone arises not only from

467

anthropogenic overexploitation but also from inherent geological constraints, including vadose zone lithology and thickness, 455 456 which play a critical role in limiting natural recharge. For example, the extended infiltration times observed at the zone's center 457 in Ningjin County, is consistent with the findings of Zhao et al. (2020). Furthermore, under equivalent vadose zone and 458 temporal conditions, riverbed infiltration, i.e., simulated with a constant head boundary, exhibited markedly greater recharge efficacy than precipitation-fed infiltration, which is in line with the studies by Nan et al. (2024). Collectively, these insights highlight the substantial benefits of implementing managed aquifer recharge via riverbed infiltration or constructing river-fed 460 infiltration basins to accelerate groundwater recovery in overexploited regions. 462 Several limitations in the current analysis point to avenues for refinement in subsequent investigations. First, the model inputs for precipitation boundary conditions started in August 2022, when the flood season was heavy, potentially biasing estimates 464 toward high-intensity events. Considering the pronounced interannual variability in precipitation patterns, differences in initial soil water content within the thick vadose zone inevitably led to variations in infiltration and recharge across different years. Future research should categorize hydrological years (e.g., wet, normal, and dry) using frequency-based analyses and simulate water flux variations under these varied regimes, thereby providing a more comprehensive understanding of infiltration time variability in the vadose zone. Second, due to the challenges in directly measuring the hydraulic parameters and actual soil water content of the thick 469 470 unsaturated zone in the study area, the current parameters were derived from the Rosetta3 model, which generalize lithological properties to some degree. Such estimations may diverge from site-specific field values, introducing potential uncertainties into the simulation results. Future work could integrate field experiments for parameter calibration and optimization, ultimately 472 improving the robustness of vadose zone modeling in similar hydrogeological settings. 473 474 Thirdly, the multiple regression models developed for percolation velocities under both precipitation-fed and riverbed 475 conditions demonstrate that the selected independent variables, including vadose zone thickness, clay content, and sand content, 476 exhibit satisfactory explanatory power. The results highlight that unsaturated zone thickness and lithological characteristics are the dominant factors controlling groundwater recharge in the Ningbailong depression zone, where thicker profiles and finer 477 textures impede percolation, while coarser sands promote faster flow. Key differences between the two infiltration conditions 478 479 include nonlinear dynamics in precipitation-fed recharge (captured by quadratic terms), suggesting diminishing slowing effects at extreme thickness or clay levels, implying some resilience in severely groundwater depression zones. In contrast, riverbed 480 recharge shows more linear patterns, enabling faster velocities and shorter delays. This underscores riverbed recharge's 482 potential to bypass constraints more effectively than precipitation. Additionally, regarding the relative contributions to total recharge volume, while precipitation-fed infiltration dominates the 484 study area (99.65%, based on the river area of 0.35% in this region), its lower average percolation velocity (26.4 cm d<sup>-1</sup>) contrasts with the higher velocity for riverbed (109.1 cm d<sup>-1</sup>). The per-unit-area recharge from riverbed is approximately 4.1 485 486 times higher. Thus, despite its small fraction, riverbed contributes a more efficient recharge approach. In near-mountain areas 487 with coarse soils, where velocities exceed 100 cm d<sup>-1</sup> for both but riverbed remains faster, strategically increasing riverbed-

like areas could boost total recharge volume, accelerating recovery in the depression zone.

Finally, due to the limited number of observational boreholes (only 24), the regression equations performed less effectively on the validation dataset, reflecting overfitting and prediction variability. Future research should expand datasets and validate with field experiments to enhance accuracy. Ultimately, these models inform adaptive strategies, such as prioritizing managed aquifer recharge in high-permeability areas to accelerate recovery.

### **5 Conclusions**

499500

This study provides a comprehensive analysis of groundwater recharge dynamics in the Ningbailong depression zone, a typical groundwater overexploited area in the North China Plain. Using HYDRUS-1D simulations informed by site-specific borehole lithology, vadose zone thicknesses, and hydro-meteorological data, we quantified infiltration times and percolation velocities under precipitation-fed and riverbed recharge sources. Empirical regression equations were derived to relate these percolation velocities to key vadose zone properties (e.g., thickness and lithology), facilitating spatial extrapolation and elucidating dominant controls on recharge efficiency in deep, heterogeneous vadose zones. The main conclusions are summarized below: 1 Precipitation recharge, while widespread, exhibits prolonged infiltration times averaging 446 days and percolation velocities of 26.4 cm d<sup>-1</sup>, leading to lower overall efficiency and delayed groundwater replenishment. In contrast, riverbed infiltration is markedly faster and more concentrated, with average recharge times of 91 days and percolation velocity of 109.1 cm d<sup>-1</sup> under equivalent lithological conditions, highlighting its superior efficacy for rapid groundwater recovery. This disparity highlights the managed aquifer recharge strategies leveraging riverbeds, particularly in overexploited regions where precipitation alone cannot balance extraction rates. 2 Recharge dynamics are profoundly modulated by vadose zone characteristics, with thicker profiles (>50 m) and finertextured soils (e.g., clays and loams) extending infiltration times up to 1,395 days (averaging 446 days but varying from 10 days in coarse-grained zones to over a thousand days in finer ones) and slowing velocities below 5 cm d<sup>-1</sup> under precipitation-fed condition, while coarser sands facilitate faster infiltration (>200 cm d<sup>-1</sup>). These thicker profiles and finetextured soils increase recharge lags, worsening groundwater depletion by decoupling surface inputs from aquifer responses, potentially delaying recovery by years. In the Ningbailong depression zone, such geological constraints and combined with historical overexploitation likely contribute to the persistence of the depression,-keeping the water supply and demand out of balance. Recharge processes display pronounced spatial variability, with infiltration times increasing and velocities decreasing 3

from the western Taihang Mountain foothills (e.g., 

plains (>1,000 days in thicker, finer lithologies). This gradient emphasizes the foothills as likely prime locations for managed aquifer recharge strategies, due to their high-permeability sediments that enable efficient percolation.

Empirical regression equations were derived through multiple regression analysis with leave-one-out cross-validation to predict percolation velocities based on vadose zone thickness (log-transformed), clay content, and sand content. The regression equations exhibit good calibration performance, with  $R^2$  values averaging around 0.7 for precipitation-fed recharge and 0.8 for riverbed recharge. However, the overall validation  $R^2$  values are modest (0.47 for both precipitation-fed and riverbed), reflecting the limited sample size (24 boreholes), which may contribute to prediction variability and potential overfitting. Expanding the dataset with additional boreholes, would potentially enhance model robustness, generalizability, and predictive accuracy. Nonetheless, these equations provide valuable preliminary insights into the influence of lithology and thickness on recharge dynamics, enabling estimates of delays and efficiencies at unsampled sites to inform groundwater management strategies in depression zones.

With respect to recharge regimes, both precipitation-fed infiltration and riverbed infiltration are feasible approaches in the southwestern part of the study area, where their efficiencies are comparable. In contrast, across most of the central and northern regions, riverbed infiltration exhibits a substantially higher efficiency than precipitation infiltration. For these areas, it is recommended to construct reservoirs or infiltration basins to capture floods and facilitate groundwater recharge via riverbed infiltration. For practical groundwater recovery management, it is essential to incorporate vadose zone lags and heterogeneity into the strategies, with these insights holding potential to inform sustainable management in analogous groundwater depression zones.

### 534 Data availability

Data will be made available on request.

#### **Author contributions**

Shenghao XU contributed to the conceptualization, data curation, formal analysis, methodology, software, visualization, and writing (original draft preparation). Yonggen Zhang contributed to the conceptualization, funding acquisition, methodology, project administration, supervision, and writing (review and editing). Xinwang Li contributed to the data curation, and investigation. Jianzhu Li contributed to the conceptualization, funding acquisition, supervision, and writing (review and editing). Wenhao Shi contributed to the software, and visualization. Shaowei Lian contributed to the data curation, and

- resources. Lei Li contributed to the data curation, and investigation. Lutz Weihermüller contributed to the methodology, and
- writing (review and editing). Marcel Schaap contributed to the methodology, and writing (review and editing).
- JY contributed to the conceptualization, methodology, formal analysis, and writing (review and editing). IH contributed to the
- writing (review and editing). TX contributed to the conceptualization and review and editing. CL contributed to the
- methodology, and review and editing.

### **Competing interests**

- The authors have the following competing interests: Yonggen Zhang is a member of the editorial board of Hydrology and
- Earth System Sciences.

### 550 Acknowledgments

- This work was supported by the National Key R&D Program of China (grant number: 2023YFC3006503) and the National
- Natural Science Foundation of China (grant numbers: 4247022276, 42077168). Schaap was supported, in part, by USDA-
- NIFA W-4188 "Soil Water and Environmental Physics to Sustain Agriculture and Natural resources" and USDA-NRCS
- Award NR233A750023C022.

### 555 Appendix A. Regression coefficients obtained from the multiple regression model with leave-one-out cross-validation.

- Depth represents the logarithm of the vadose zone thickness (m), clay represents clay fraction (-), and sand represents sand
- fraction (-).  $R^2$  cal and  $R^2$  val indicate the coefficient of determination for the calibration and validation datasets.

Table A.1. Regression coefficient of the multiple regression model for the logarithm of the average percolation velocity (cm d<sup>-1</sup>) under precipitation recharge conditions.

| Set | Intercept | Depth  | Clay  | Sand  | Depth*Clay | Depth*Sand | Depth <sup>2</sup> | Clay <sup>2</sup> | R <sup>2</sup> _cal | R <sup>2</sup> _val |
|-----|-----------|--------|-------|-------|------------|------------|--------------------|-------------------|---------------------|---------------------|
| 1   | 24.86     | -15.77 | 54.10 | 14.09 | -14.84     | -3.63      | 2.60               | 6.79              | 0.71                |                     |
| 2   | 24.28     | -15.49 | 54.33 | 14.25 | -14.91     | -3.67      | 2.57               | 6.86              | 0.71                |                     |
| 3   | 28.18     | -17.29 | 52.78 | 12.72 | -14.46     | -3.27      | 2.77               | 6.69              | 0.72                |                     |
| 4   | 21.83     | -13.32 | 50.61 | 11.03 | -13.79     | -2.80      | 2.16               | 6.11              | 0.66                |                     |
| 5   | 24.89     | -15.77 | 53.98 | 14.04 | -14.81     | -3.61      | 2.60               | 6.80              | 0.70                |                     |
| 6   | 8.53      | -8.16  | 70.14 | 17.19 | -18.82     | -4.41      | 1.72               | 6.74              | 0.73                | 0.47                |
| 7   | 21.07     | -14.50 | 53.60 | 19.50 | -14.64     | -5.07      | 2.53               | 6.29              | 0.74                | 0.47                |
| 8   | 30.44     | -18.57 | 53.01 | 13.57 | -15.01     | -3.57      | 2.97               | 8.59              | 0.83                |                     |
| 9   | 26.99     | -16.55 | 54.57 | 11.43 | -15.02     | -2.93      | 2.66               | 7.26              | 0.71                |                     |
| 10  | 27.20     | -16.80 | 53.14 | 12.95 | -14.61     | -3.32      | 2.71               | 7.04              | 0.71                |                     |
| 11  | 18.38     | -12.90 | 54.93 | 17.99 | -15.05     | -4.59      | 2.29               | 6.62              | 0.72                |                     |
| 12  | 22.61     | -14.71 | 59.32 | 12.76 | -16.11     | -3.29      | 2.47               | 6.58              | 0.71                |                     |

| Set                   | Intercept | Depth  | Clay  | Sand  | Depth*Clay | Depth*Sand | Depth <sup>2</sup> | Clay <sup>2</sup> | R <sup>2</sup> _cal | R <sup>2</sup> _val |
|-----------------------|-----------|--------|-------|-------|------------|------------|--------------------|-------------------|---------------------|---------------------|
| 13                    | 30.63     | -17.44 | 51.23 | 6.90  | -14.28     | -1.88      | 2.67               | 7.53              | 0.71                |                     |
| 14                    | 20.30     | -14.54 | 55.91 | 20.02 | -15.34     | -5.12      | 2.58               | 7.14              | 0.73                |                     |
| 15                    | 24.74     | -15.72 | 54.08 | 14.20 | -14.84     | -3.66      | 2.60               | 6.80              | 0.70                |                     |
| 16                    | 36.26     | -21.60 | 58.03 | 13.70 | -15.77     | -3.51      | 3.34               | 6.56              | 0.52                |                     |
| 17                    | 24.29     | -15.43 | 55.41 | 13.41 | -15.28     | -3.45      | 2.55               | 7.43              | 0.71                |                     |
| 18                    | 21.57     | -14.32 | 55.18 | 15.73 | -15.06     | -4.06      | 2.45               | 6.29              | 0.71                |                     |
| 19                    | 21.23     | -14.12 | 55.36 | 15.46 | -15.11     | -3.97      | 2.42               | 6.31              | 0.72                |                     |
| 20                    | 39.71     | -22.33 | 43.45 | 7.49  | -11.83     | -1.89      | 3.29               | 5.33              | 0.70                |                     |
| 21                    | 29.08     | -17.53 | 49.19 | 11.92 | -13.30     | -3.02      | 2.76               | 5.35              | 0.70                |                     |
| 22                    | 28.09     | -17.22 | 50.59 | 13.04 | -14.37     | -3.39      | 2.77               | 11.34             | 0.71                |                     |
| 23                    | 28.21     | -17.33 | 52.94 | 12.88 | -14.63     | -3.32      | 2.78               | 7.42              | 0.72                |                     |
| 24                    | 23.82     | -15.24 | 54.62 | 14.20 | -14.97     | -3.67      | 2.54               | 6.72              | 0.70                |                     |
| Average Value         | 25.30     | -15.94 | 54.19 | 13.77 | -14.87     | -3.55      | 2.62               | 6.94              | /                   | /                   |
| Standard<br>Deviation | 6.10      | 2.80   | 4.59  | 3.07  | 1.19       | 0.78       | 0.33               | 1.16              | /                   | /                   |

Table A.2. Regression coefficient of the multiple regression model for the logarithm of the average percolation velocity (cm d<sup>-1</sup>) under riverbed recharge conditions.

| Set | Intercept | Depth | Clay   | Sand  | Depth*Clay | Depth*Sand | Clay*Sand | R <sup>2</sup> _cal | R <sup>2</sup> _val |
|-----|-----------|-------|--------|-------|------------|------------|-----------|---------------------|---------------------|
| 1   | -0.56     | 1.42  | -43.51 | 5.76  | 10.46      | -1.95      | 8.86      | 0.77                |                     |
| 2   | -0.63     | 1.46  | -43.50 | 5.29  | 10.37      | -1.88      | 9.74      | 0.78                |                     |
| 3   | -1.23     | 1.64  | -44.07 | 6.77  | 10.45      | -2.27      | 9.80      | 0.81                |                     |
| 4   | -2.31     | 1.91  | -41.02 | 7.23  | 9.72       | -2.39      | 9.68      | 0.78                |                     |
| 5   | 1.34      | 0.92  | -45.57 | 3.32  | 10.98      | -1.35      | 9.42      | 0.80                |                     |
| 6   | -1.17     | 1.58  | -36.49 | 4.96  | 8.64       | -1.79      | 10.11     | 0.80                |                     |
| 7   | -0.99     | 1.56  | -43.97 | 6.35  | 10.48      | -2.17      | 9.81      | 0.78                |                     |
| 8   | -0.28     | 1.37  | -45.09 | 5.24  | 10.74      | -1.87      | 10.53     | 0.79                |                     |
| 9   | 1.82      | 0.79  | -40.51 | 0.31  | 9.77       | -0.53      | 8.40      | 0.77                |                     |
| 10  | 0.07      | 1.26  | -43.63 | 4.44  | 10.44      | -1.64      | 9.54      | 0.79                |                     |
| 11  | 1.95      | 0.82  | -39.42 | -1.16 | 9.39       | -0.32      | 9.59      | 0.87                | 0.47                |
| 12  | -0.47     | 1.46  | -66.58 | 15.43 | 16.05      | -4.46      | 10.09     | 0.81                |                     |
| 13  | -5.29     | 2.57  | -40.34 | 12.39 | 9.69       | -3.60      | 9.32      | 0.79                |                     |
| 14  | 0.65      | 1.13  | -43.80 | 3.68  | 10.47      | -1.47      | 9.52      | 0.78                |                     |
| 15  | -1.64     | 1.72  | -44.08 | 7.29  | 10.58      | -2.40      | 9.27      | 0.78                |                     |
| 16  | 1.77      | 0.84  | -56.79 | 1.74  | 13.76      | -0.98      | 9.41      | 0.76                |                     |
| 17  | -0.64     | 1.46  | -44.27 | 5.79  | 10.55      | -2.02      | 9.90      | 0.78                |                     |
| 18  | -0.63     | 1.46  | -43.59 | 5.53  | 10.40      | -1.95      | 9.67      | 0.79                |                     |
| 19  | -0.32     | 1.37  | -43.36 | 5.03  | 10.37      | -1.81      | 9.47      | 0.79                |                     |
| 20  | -1.14     | 1.59  | -41.73 | 6.20  | 9.97       | -2.11      | 9.14      | 0.76                |                     |
| 21  | -1.00     | 1.55  | -41.00 | 5.77  | 9.82       | -2.00      | 8.73      | 0.77                |                     |

| Set                | Intercept | Depth | Clay   | Sand | Depth*Clay | Depth*Sand | Clay*Sand | R <sup>2</sup> _cal | R <sup>2</sup> _val |
|--------------------|-----------|-------|--------|------|------------|------------|-----------|---------------------|---------------------|
| 22                 | -1.51     | 1.81  | -44.51 | 6.74 | 9.63       | -2.40      | 15.30     | 0.80                |                     |
| 23                 | -0.48     | 1.42  | -43.62 | 5.33 | 10.41      | -1.89      | 9.67      | 0.78                |                     |
| 24                 | -0.39     | 1.40  | -43.47 | 5.14 | 10.38      | -1.85      | 9.57      | 0.78                |                     |
| Average Value      | -0.54     | 1.44  | -44.33 | 5.61 | 10.56      | -1.96      | 9.77      | /                   | /                   |
| Standard Deviation | 1.51      | 0.38  | 5.91   | 3.31 | 1.47       | 0.84       | 1.26      | /                   | /                   |

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
