# Peer review of "Quantification of Delayed Recharge by Soil Surface and Riverbed"

_EGUsphere, 2025_

## Author Comment (AC1)

**Re: Revised Manuscript Preprint egusphere-2025-5651 (Quantification of Delayed Recharge by Soil Surface and Riverbed Infiltration in a Deep Groundwater Depression Zone in the North China Plain)**

**Authors' Responses to Comments from Dr. Nima Zafarmomen:**

**General Comments**

*The paper addresses an important and very topical problem: delayed recharge in deep vadose zones within a major groundwater depression cone in the North China Plain, comparing precipitation-fed vs riverbed recharge using HYDRUS-1D plus borehole lithology. The regional perspective and explicit focus on lag times and percolation velocities are valuable and fit well within hydrology / groundwater journals. I recommend it for publication after considering below comments.*

**Response**: We sincerely thank Dr. Nima Zafarmomen for the positive evaluation and the encouraging summary of our work. We appreciate your recognition of the importance of this study regarding delayed recharge in the deep vadose zones of the North China Plain. We are also pleased that the regional perspective and the specific focus on lag times and percolation velocities using HYDRUS-1D and borehole lithology were well-received. We have carefully addressed all specific comments and will incorporate the necessary revisions into the final manuscript to enhance its clarity and scientific rigor.

**Specific Comments**

*1. You currently equate "recharge efficiency" mostly with higher percolation velocity and shorter lag time, but sometimes imply it means larger recharge volume. Please give a clear, formal definition early in the paper and stick to it. When you say riverbed recharge is ~4.1× higher "per unit area", clarify this is based on velocity, not on simulated recharge flux volume, or explicitly compute and show fluxes.*

**Response 1**: We sincerely thanks for pointing out the ambiguity regarding the term "recharge efficiency". We agree that a formal definition is necessary to distinguish between the rapidity of the process and the total recharge volume. To address this, we will implement the following modifications in the revised manuscript:

1) We will add a clear definition of "recharge efficiency" in the Introduction section. We will explicitly define it as the rapidity of the vadose zone response, quantified by average percolation velocity rather than the total volume of water.

2) In the Discussion section, we will clarify that the statement "4.1 times higher" refers to the recharge rate based on average percolation velocities, not the total simulated flux volume.

*2. Key modeling choices—uniform initial head (–50 cm), 1D vertical flow only, and no root uptake for riverbed cases—are reasonable but need clearer justification. Explain that the long spin-up minimizes sensitivity to the initial profile and that omitting ET in riverbeds makes the riverbed scenario optimistic. Also acknowledge that lateral flow, preferential flow, and riverbed clogging are not represented and discuss qualitatively how this may bias lag times.*

**Response 2:** Thank you for highlighting the need to better justify our modeling assumptions and discuss their implications. We agree that while these simplifications are standard for regional-scale vadose zone modeling, their potential biases should be explicitly addressed. We will revise the manuscript in the "Methods" and "Discussion" sections to address these points and these additions will provide a balanced and transparent interpretation of the model's capabilities and limitations. The following clarifications and justifications will be incorporated into the revised manuscript:

1) We will clarify that the 6-year spin-up period (2016-2022) is specifically implemented to minimize the sensitivity of the simulation results to the uniform initial pressure head (-50 cm). This ensures that the soil water profile reached dynamic equilibrium before the analysis period. We will add a new figure (Figure B1 in the Appendix B.) that displays the temporal evolution of soil water content at deep layers (20-80 m) for all boreholes during the spin-up phase.

2) We will add an explanation that neglecting root water uptake in the riverbed scenario provides an upper bound estimate of recharge efficiency.

3) We will add a new paragraph in the Discussion section to qualitatively analyze the biases introduced by 1-D flow and preferential flow, and riverbed clogging.

Regarding riverbed clogging, our model setup was based on detailed borehole lithology which was parameterized into seven categories (as described in Section 2.3.1). This detailed parameterization explicitly included low-permeability layers (such as clays and silts) at various depths, and to a certain extent, it can reflect the obstructive effect of low-permeability layers on riverbed infiltration.

*3. IDW interpolation of 24 points over ~2,000 km² is appropriate for a first-order picture but provides no uncertainty and may be weak where points are sparse. Clarify that maps of infiltration time and velocity should be interpreted qualitatively, especially in poorly constrained regions. Briefly justify the choice of IDW over kriging (e.g., limited data for robust variogram fitting) and mention this as a limitation.*

**Response 3:** We appreciate the reviewer's insightful comment regarding the spatial interpolation method. We acknowledge that with a limited dataset (num = 24), the resulting maps serve primarily as a regional trend. We will revise the manuscript to address these points explicitly, including:

1)  We will add a justification for selecting IDW over Kriging and explain that the limited number of data points was insufficient for robust variogram fitting, making IDW a more appropriate choice for approximating general trends in this context.

2)  In the Results section, where the maps are introduced, we will add a cautionary note stating that the maps should be interpreted qualitatively, especially in regions with sparse data coverage.

3)  In the Discussion section, we will explicate that the spatial analysis was constrained by the sparsity of points and the lack of uncertainty quantification in the IDW method.

*4. The constant-head lower boundary at the long-term average groundwater level is a strong simplification in a declining groundwater system and likely underestimates true lag times. Justify this assumption more clearly and discuss its effect on results; a short sensitivity discussion would help. Similarly, using a single rainfall station and a single river stage series for the whole area needs explicit justification and acknowledgement of added uncertainty.*

**Response 4:** We sincerely thanks for identifying these critical simplifications regarding the boundary conditions and forcing data. We agree that these assumptions require explicit justification and a discussion of their implications. We will revise the manuscript to address these concerns as follows:

1) Considering that the actual groundwater level in the North China Plain is constantly changing, we agree that assuming a constant groundwater level is a simplification. We will add a discussion in the Discussion section acknowledging the sensitivity of the results to this assumption. In the revised manuscript, we will acknowledge that the groundwater level dynamics are complex, with possibilities for both decline (due to extraction) and rise (due to management), and we will continue to optimize and solve this problem in subsequent studies.

2) We will add text to explicitly justify the use of single-station data. As stated in the revised manuscript, the Baixiang Rain Gauge Station and the selected river stage sequence were chosen for the high continuity of their observational records within the study area. Furthermore, applying these data uniformly across the region serves a specific methodological purpose, i.e., to control meteorological variables. By keeping the surface inputs constant, we can isolate and focus primarily on the influence of vadose zone heterogeneity (thickness and lithology) on infiltration recharge, which is the central objective of this study.

*5. I strongly recommend to discuss the paper and deepen your discussion "Assimilation of sentinel‑based leaf area index for modeling surface‑ground water interactions in irrigation districts".*

**Response 5:** Thank you for the comments. We have carefully reviewed the recommended paper and agree that it offers critical insights into improving the representation of vegetation dynamics in hydrological modeling. We will integrate a discussion of this work into the Discussion section of the revised manuscript.

*6. The phrase "two infiltration modes were considered: precipitation-fed and riverbed infiltration" could be tightened to "precipitation-fed soil infiltration and riverbed infiltration".*

**Response 6:** Thank you for the comments. We will modify the phrase in the Abstract exactly as suggested.

*7. When mentioning the regression equations, briefly state the key predictors (vadose zone thickness and particle fractions) to give the reader more context.*

**Response 7:** Thank you for the comments. We will revise the sentence to explicitly list the specific independent variables used in the regression analysis.

*8. Some paragraphs are quite long and dense (e.g., lines 41–64, 85–110). Consider splitting into shorter paragraphs to improve readability.*

**Response 8:** Thank you for the comments. Clarifications will be made in lines 41-64 and 85-110, Introduction of the revised manuscript.

*9. When you review past work (HYDRUS applications, global lag studies), explicitly state the remaining gap you are addressing (combined effect of deep vadose zones, complex lithology, and 'comparison of two recharge sources under identical profiles)'. You do this, but it could be more sharply framed at the end of the Introduction.*

**Response 9:** Thank you for the comments. We will rewrite the paragraph at the end of the Introduction to explicitly state the gap regarding the combined effects of lithology and depth, and the lack of comparisons under identical profiles.

*10. It might be helpful to explicitly mention average annual precipitation and reference ET, if available, to characterize the climate quantitatively.*

**Response 10:** Thank you for the comments. We will add the long-term average precipitation and evaporation data to Section 2.1 Study Area.

*11. The description of boundaries (Taihang Mountains, Shijiazhuang, Hengshui) is good, but consider adding one sentence stating dominant land use (e.g., double cropping, wheat–maize rotation) to connect with the root uptake assumptions.*

**Response 11:** Thank you for the comments. We will add a description of the dominant cropping system to Section 2.1 (Study Area)

*12. "Depth (cm)" is given for boreholes, but values like 8,080 cm (= 80.8 m) etc. Make clear that these are vadose zone thicknesses down to shallow groundwater table or borehole depth; the phrase "Depth (cm)" is ambiguous.*

**Response 12:** Thank you for the comments. We will clarify that the values listed in Table 1 represent the thickness of the vadose zone rather than the total depth of the borehole and will add an explanatory note to the caption of Table 1.

*13. You might add a column indicating vadose zone thickness vs. total borehole depth if they differ.*

**Response 13:** Thank you for the comments. Since the focus of this study is on vadose zone infiltration and recharge, the "Vadose Zone Thickness" is the critical vertical parameter. As detailed in our response to the previous comment, we have explicitly clarified in the caption of Table 1 that the listed "Depth" refers specifically to the vadose zone thickness. We believe this clarification effectively removes the ambiguity regarding the vertical dimension used in our models without needing an additional column.

---

## Author Comment (AC2)

**Authors' Responses to Comments from Reviewer #1:**

**General Comments**

*In this paper, the authors present a solid and well-structured study addressing an important issue in groundwater sustainability in the North China Plain. The use of HYDRUS-1D to analyze recharge delays in thick vadose zones is appropriate and well justified. In addition, the comparison of recharge efficiency and time lag under identical vadose zone conditions for two distinct recharge modes is novel and provides clear practical relevance for managed aquifer recharge (MAR) planning in groundwater depression zones. The results, as currently presented, also seem reasonable. I encourage the authors to further extend their analysis to address several concerns and questions that other readers may also raise regarding the flexibility of the approach and the range of conditions under which the method can be reliably applied. My main concerns are summarized as follows.*

**Response**: We sincerely thank the reviewer for the encouraging evaluation of our work. We heavily appreciate your recognition of the study's structure, the appropriateness of the HYDRUS-1D modeling approach, and the novelty of comparing recharge efficiencies between precipitation-fed and riverbed infiltration.

We also value your constructive suggestion to further extend the analysis regarding the flexibility and applicability range of our method. We have carefully considered your specific concerns below and will incorporate additional analyses and discussions into the revised manuscript to address the robustness of our approach under varying conditions. Please see our point-by-point responses below.

**Specific Comments**

*1. The authors clearly highlight heterogeneity in horizontal boundary conditions across the North China Plain (Table 1), where some locations are characterized by precipitation–evaporation–infiltration processes, while others are governed by constant-head riverbed infiltration. At the same time, strong vertical heterogeneity in soil types is emphasized (Figure 2). Given these heterogeneities, it is not fully clear whether a purely one-dimensional modeling framework can adequately resolve the*

*dominant flow processes. For example, when infiltrating water encounters low-permeability layers at depth, lateral flow along stratigraphic interfaces may occur. Such lateral redistribution could potentially influence infiltration times and recharge efficiency. Under these heterogeneous horizontal boundary conditions, lateral soil water flow may not be negligible, unlike in traditional large-scale studies where lateral flow is often assumed to be insignificant. The manuscript would therefore benefit from a discussion of the potential magnitude of lateral flow and its implications, as neglecting horizontal flow may limit the applicability of a one-dimensional approach in settings with heterogeneous boundary conditions.*

**Response 1:** We sincerely thank the reviewer for this insightful comment regarding the dimensionality of our modelling approach. We fully agree that lateral redistribution, particularly in heterogeneous domains, is a critical process that is simplified in our 1D framework. To address this, we will add a dedicated paragraph in the Discussion section to explicitly analyse the implications and potential magnitude of this simplification, supported by comparative literature between 1D and 2D models.

We acknowledged that neglecting lateral flow likely leads to an underestimation of lag times compared to reality, as lateral flow extends the travel path (Isch et al., 2022). We incorporated findings from Chen et al. (2022) to provide a quantitative reference for the potential error. Their simulations in heterogeneous domains suggested that neglecting lateral connectivity could lead to peak flow deviations of approximately 8.0% and timing shifts of 4 to 17 minutes.

We clarified that while 2-D models are superior for specific conditions (e.g., furrow irrigation as noted by Crevoisier et al., 2008), they strictly require detailed data on horizontal stratigraphic continuity, which is unavailable at our regional scale. Forcing a 2D model without such data would introduce greater uncertainty. On the other hand, previous studies in the North China Plain (Huo et al., 2014) have demonstrated that vertical flow remains the dominant mechanism for groundwater recharge in this thick vadose zone.

We believe this expanded discussion will provide a balanced view of the model's limitations and justifies the applicability of HYDRUS-1D for the study's objectives.

*2. Equations (2) – (6) describe the van Genuchten–Mualem constitutive relationships. However, the formulation appears inconsistent in places, and some parameters (e.g., $\theta_s$ and $\theta_r$) are not clearly defined when first introduced. Clarifying the parameter definitions and ensuring consistency with standard van Genuchten notation would improve transparency and reproducibility.*

**Response 2:** Thank you for checking the mathematical formulation. We utilized the modified van Genuchten-Mualem model as proposed by Vogel et al. (2000), which is implemented in HYDRUS-1D to improve numerical convergence near saturation by introducing a small air-entry pressure head ($h_s$). We will revise Section 2.3.2 to present the complete set of equations for these modified equations (Equations 2-7 in the revised manuscript). Furthermore, as requested, we will explicitly define all parameters (including $\theta_r$, $\theta_s$, $\theta_m$, $h_s$, et al.) immediately following their introduction to ensure clarity and reproducibility.

*3. The terms "infiltration time" or "recharge time" are used throughout the manuscript. I recommend explicitly defining these quantities early in the Methods section, preferably in mathematical form, and using the terminology consistently thereafter to avoid ambiguity.*

**Response 3:** Thank you for the comments. We will describe these definitions in Section 2.4 (Spatial Interpolation), and we will standardize the terminology throughout the manuscript. We now consistently use the term "infiltration time" to refer to the lag time, replacing the previous mix of "recharge time" and "infiltration time."

---

## Author Comment (AC3)

**Authors' Responses to Comments from Reviewer #2:**

**General Comments**

*This manuscript investigates the mechanisms of soil water movement and groundwater recharge in the North China Plain, a region facing significant water scarcity issues. The authors employ numerical simulations combined with multiple regression analysis to quantify the influence of vadose zone thickness, soil texture (clay and sand fractions), and lithology on percolation velocities and recharge rates. The study compares different infiltration modes (such as precipitation vs. managed aquifer recharge/riverbed infiltration) and aims to provide theoretical support for sustainable groundwater extraction and crop production. The topic is highly relevant to regional water resources management and addresses a pressing global concern regarding aquifer depletion. But there are significant methodological concerns that must be addressed to ensure the validity of the results.*

**Response**: We sincerely thank the reviewer for the comprehensive summary and for recognizing the high relevance of our study to regional water resources management in the North China Plain and the global concern of aquifer depletion. We greatly appreciate your positive evaluation of our research topic and objectives.

In the revised manuscript, we will address these issues to ensure the validity and robustness of our results. Specifically, we will strengthen the justification for our model parameterization and improved the statistical rigor of our regression analysis.

*1. Since measured initial soil water content was unavailable, the reliability of the simulation results depends heavily on the sufficiency of the spin-up period. It is unclear from the current text whether the spin-up period has made the initial state reached a dynamic equilibrium state, especially for deep soil layers. The authors should provide justification or graphical evidence (e.g., time series of soil water content or pressure head deeper than a certain depth at the profile) to demonstrate that the model achieved a robust equilibrium prior to the main simulation period.*

**Response 1:** We thank the reviewer for this critical comment regarding the initial conditions and the model spin-up. We fully agree that in the absence of measured initial

profiles for such deep vadose zones, ensuring a sufficient spin-up period is essential to eliminate the influence of the arbitrary initial setup (uniform pressure head of -50 cm). In the revised manuscript, we will provide additional graphical evidence to justify the sufficiency of the 6-year spin-up period (July 2016 - July 2022). Specifically:

1) We will add a new figure (Figure B1 in the Appendix B.) that displays the temporal evolution of soil water content at deep layers (at intervals of 10 m from 20 m to 80m) for all boreholes during the spin-up phase.

2) We will update Section 2.3.5 (Model spin-up) to include this analysis, clarifying that the system had achieved a robust equilibrium prior to the main simulation period starting in August 2022.

*2.The multiple regression models employ clay, sand fractions, and depth as independent predictors. Since soil textural components are compositional data (summing to 100% with silt, clay and sand), there is an inherent negative correlation between these variables. To ensure the robustness of the regression coefficients presented in Tables A.1 and A.2, please check the independence of the input variables or calculate the Variance Inflation Factors for the predictors. If high multicollinearity is detected, the authors should discuss how this affects the physical interpretation of the coefficients.*

**Response 2:** We thank the reviewer for raising this important statistical point. To verify the independence of the selected predictors (vadose zone thickness, clay fraction, and sand fraction), we will perform a multicollinearity test using the Variance Inflation Factor (VIF). In our preliminary analysis, the VIF values for all three predictors have been calculated to be well below the critical threshold of 5 (1.018 for Depth, 1.208 for Clay, and 1.225 for Sand), indicating that there is no significant multicollinearity among the predictors in our dataset. We will add these VIF values to Section 2.5 Multiple regression analysis to confirm the independence of the input variables.

*3. The Conclusions suggest the construction of recharge basins; however, the implications for groundwater management could be further strengthened. Given the*

*limited land resources in the North China Plain, does this recommendation adequately consider land-use constraints?*

**Response 3:** We appreciate the reviewer's valuable perspective on land-use constraints. We are fully aware that the North China Plain is a densely populated region and a critical agricultural base where land resources are extremely limited. We will refine our recommendations in the Discussion and Conclusion sections to explicitly address land scarcity. Instead of proposing the conversion of arable land into large-scale artificial basins, we advocate for a strategy that optimizes Managed Aquifer Recharge (MAR) by leveraging existing geomorphological features.

We will clarify that recharge basins should be prioritized along existing seasonal dry riverbeds or river edges. As noted in our study area description, the main river covers only 0.35% of the area, and many channels are seasonally dry. Utilizing these channels minimizes land acquisition costs and avoids conflict with agricultural land use.

We will emphasize our finding that riverbed percolation velocities (109.1 cm d$^{-1}$) are approximately 4 times higher than precipitation-fed infiltration (26.4 cm d$^{-1}$). This high efficiency implies that a smaller surface area can achieve significant recharge volumes, reducing the need for extensive land occupation.

We specifically recommended targeting the foothill alluvial fan areas (e.g., near the Long 12 and Bai 1), where coarse-grained lithology combined with riverbed infiltration yields the shorter lag times.

**Specific comments:**

*1.Line 18 "...sustainable groundwater extraction and crop-production," "Crop production" is typically not hyphenated unless used as a compound modifier before a noun.*

**Response 1:** We appreciate these suggestions. Clarifications will be made on Page 1, in the Abstract of the revised manuscript.

*2. Line 19, The word "however" is redundant following "few studies".*

**Response 2:** We appreciate these suggestions. Clarifications will be made on Page 1, in the Abstract of the revised manuscript.

*3.Line 23 "…compared between these two infiltrations," the word "infiltrations" is a non-standard pluralization in this context. Please revise "compared between these two infiltrations" to "compared between these two infiltration modes".*

**Response 3:** We appreciate these suggestions. Clarifications will be made on Page 1, in the Abstract of the revised manuscript.

*4. Line 33, there is a subject-verb agreement error. "…the depletion of aquifers have become…" should be corrected, as the subject "depletion" is singular.*

**Response 4:** We appreciate these suggestions. Clarifications will be made on Page 2, in the Introduction of the revised manuscript.

*5. Line 33, in the same sentence, "pressing global concerns" should be changed to "a pressing global concern" to match the singular subject.*

**Response 5:** We appreciate these suggestions. Clarifications will be made on Page 2, in the Introduction of the revised manuscript.

*6. Line 43, In soil science, the standard term is "matric potential." Please change "matrix potential gradient" to "matric potential gradients".*

**Response 6:** We appreciate these suggestions. Clarifications will be made on Page 2, in the Introduction of the revised manuscript.

*7. Line 118, A preposition is missing in the phrase "holistic understanding soil water movement." Please change it to "holistic understanding of soil water movement".*

**Response 7:** We appreciate these suggestions. Clarifications will be made on Page 4, in the Introduction of the revised manuscript.

*8. Line 198, it should be corrected to "and t represents time".*

**Response 8:** We appreciate these suggestions. Clarifications will be made on Page 9, Section 2.3 of the revised manuscript.

*9. Line 202, "set up" should be written as two words when used as a verb.*

**Response 9:** We appreciate these suggestions. Clarifications will be made on Page 9, Section 2.3 of the revised manuscript.

*10. Line 244, in the sentence "…and then apply the same average water level data…" the verb "apply" should be in the past tense to match the preceding "were obtained".*

**Response 10:** We appreciate these suggestions. Clarifications will be made on Page 11, Section 2.3.3 of the revised manuscript.

*11. Line 328, for more formal academic tone, please change "interplay between these factors in controlling water movement" to "interaction between these factors in governing water movement."*

**Response 11:** We appreciate these suggestions. Clarifications will be made on Page 16, Section 3.2.1 of the revised manuscript.

*12. Line 335, to ensure grammatical parallelism with "thickness," please change the adjective "lithological" to the noun "lithology".*

**Response 12:** We appreciate these suggestions. Clarifications will be made on Page 17, Section 3.2.1 of the revised manuscript.

*13. The fonts in Figures 5 and 8 are inconsistent with those in the manuscript. If there is no special meaning, it is recommended to make them consistent with the fonts of the other figures.*

**Response 13:** We appreciate these suggestions. Clarifications will be made in Figures 5 and 8 of the revised manuscript.

*14. In the titles of Figures 5 and 8, the unit of "Groundwater recharge time" should be marked as (d).*

**Response 14:** We appreciate these suggestions. Clarifications will be made in the titles of Figures 5 and 8 of the revised manuscript.

"Figure 5: Groundwater infiltration time (d) and average percolation velocity (cm $d^{-1}$) for locations under precipitation infiltration recharge scenarios."

"Figure 8: Groundwater infiltration time (d) and average percolation velocity (cm d$^{-1}$) for locations under riverbed infiltration recharge scenarios."